# Metagenome-Assembled Viral Genomes Analysis Reveals Diversity and Infectivity of the RNA Virome of Gerbillinae Species

**DOI:** 10.3390/v14020356

**Published:** 2022-02-09

**Authors:** Han Du, Lijuan Zhang, Xinqiang Zhang, Fengze Yun, Yuhao Chang, Awaguli Tuersun, Kamila Aisaiti, Zhenghai Ma

**Affiliations:** Xinjiang Key Laboratory of Biological Resources and Genetic Engineering, College of Life Science and Technology, Xinjiang University, Urumqi 830046, China; khanhtoe@icloud.com (H.D.); lijuancheung@outlook.com (L.Z.); zhangxinqiang0106@outlook.com (X.Z.); Lyber0096@outlook.com (F.Y.); a1574592121@outlook.com (Y.C.); awagulxj@outlook.com (A.T.); kamilaasat@outlook.com (K.A.)

**Keywords:** rodent, viral metagenomic, RNA virome, zoonotic viruses, *Flaviviridae*, *Picobirnaviridae*

## Abstract

Rodents are a known reservoir for extensive zoonotic viruses, and also possess a propensity to roost in human habitation. Therefore, it is necessary to identify and catalogue the potentially emerging zoonotic viruses that are carried by rodents. Here, viral metagenomic sequencing was used for zoonotic virus detection and virome characterization on 32 Great gerbils of *Rhombomys opimus*, *Meriones meridianus*, and *Meiiones Unguiculataus* species in Xinjiang, Northwest China. In total, 1848 viral genomes that are potentially pathogenic to rodents and humans, as well as to other wildlife, were identified namely *Retro-*, *Flavi-*, *Pneumo-*, *Picobirna-*, *Nairo-*, *Arena-*, *Hepe-*, *Phenui-*, *Rhabdo-*, *Calici-*, *Reo-*, *Corona-*, *Orthomyxo-*, *Peribunya-,* and *Picornaviridae* families. In addition, a new genotype of rodent Hepacivirus was identified in heart and lung homogenates of seven viscera pools and phylogenetic analysis revealed the closest relationship to rodent Hepacivirus isolate RtMm-HCV/IM2014 that was previously reported to infect rodents from Inner Mongolia, China. Moreover, nine new genotype viral sequences that corresponded to *Picobirnaviruses* (PBVs), which have a bi-segmented genome and belong to the family *Picobirnaviridae*, comprising of three segment I and six segment II sequences, were identified in intestines and liver of seven viscera pools. In the two phylogenetic trees that were constructed using ORF1 and ORF2 of segment I, the three segment I sequences were clustered into distinct clades. Additionally, phylogenetic analysis showed that PBV sequences were distributed in the whole tree that was constructed using the RNA-dependent RNA polymerase (RdRp) gene of segment II with high diversity, sharing 68.42–82.67% nucleotide identities with other genogroup I and genogroup II PBV strains based on the partial RdRp gene. By RNA sequencing, we found a high degree of biodiversity of *Retro-*, *Flavi-*, *Pneumo-,* and *Picobirnaridae* families and other zoonotic viruses in gerbils, indicating that zoonotic viruses are a common presence in gerbils from Xinjiang, China. Therefore, further research is needed to determine the zoonotic potential of these viruses that are carried by other rodent species from different ecosystems and wildlife in general.

## 1. Introduction

The *Rodentia* order, which possesses strong ecological adaptability and strong reproductive ability, is the largest mammalian order with approximately 33 families and ca 2277 living species (~40% of all mammalian species) that are distributed widely throughout the world [1,2,3,4]. Many rodents have mixed diets, and approximately 90% of them carry more than 200 pathogens that could transmit more than 60 known zoonoses globally [5,6]. They are extremely diverse in their ecology and lifestyles and can be found in almost every terrestrial habitat, including human-made environments that offer numerous opportunities for cross-species viral transmission through their urine, feces, or their arthropod ectoparasites such as ticks, mites, and fleas [7,8,9].

Continuous scientific monitoring has revealed that the rodents are the host to an ever-expanding community of zoonotic viruses including members of the family *Flaviviridae*, *Rhabdoviridae*, *Reoviridae*, *Arenaviridae*, *Picobirnaviridae*, *Nairoviridae*, *Picornaviridae*, and *Hantaviridae*, among others [2,10,11,12]. Recent metagenomic studies have identified a wide diversity of known and novel viruses in rodents from families or genera that contain important zoonotic viruses, such as the new species or variants of picornaviruses, Lassa virus, Hantaan virus (HTNV), hepatitis C virus (HCV) [13], tick-borne encephalitis virus (TBEV) [14], lymphocytic choriomeningitis virus (LCMV) [15], Moloney murine sarcoma virus (M-MSV), murine leukemia virus (MuLV), hepatitis A, and hepatitis E, etc. All of which are highly related to human infectious diseases [16,17]. Hantaviruses are a family of viruses that are transmitted mainly through direct contact with the feces, saliva, or urine of infected rodents or by inhalation of the virus in their aerosolized excreta [18,19]. Therefore, Hantavirus infection to humans is considered a spillover infection that can cause two main types of serious illnesses namely Haemorrhagic Fever with Renal Syndrome (HFRS) and Hantavirus Pulmonary Syndrome (HPS) [20,21,22,23]. HCV, with a genome of approximately 9600 nucleotides (nts) that are composed of one long ORF, is a member of the genus *Hepacivirus*, which is one of the four genera within the Flaviviridae family (+ssRNAs) [24]. The origin of viruses for many emerging diseases remains elusive, as is HCV [25], which was once an isolated infection in humans and non-human primates, but has now been expanded to include horses [26], rodents [17,27,28], bats [29], cows [30], and other wildlife [31,32,33,34]. Such evidence further reveals the wide host adaptation and the high potential for cross-species transmission of Hepacivirus and paves the way for further identification of other primate hepaciviruses. Picobirnaviruses (PBVs), the only genus in the *Picobirnaviridae* viral family, are small, non-enveloped viruses with bi-segmented double-stranded RNA genomes. Initially found in rat intestines, PBV has since been found in the fecal matter of numerous mammals with and without disease worldwide [35,36,37,38,39,40]. That many emerging infectious diseases (EIDs) are zoonoses, as mentioned above, emphasizes the importance of the species barrier in preventing transmission of infectious diseases [41], and several recent examples that illustrate the potentially disastrous consequences that can occur when the barrier is breached [42,43].

It is impossible to estimate the species and number of viruses that are transmitted both within and between species throughout the life of rodents that harbor a vast number of unknown viruses [6,17,44]. Thus, the comprehensive understanding of viral community, as well as the prevalence, genetic diversity, and geographical distribution of these viruses, especially the unknown ones that are present in rodents and other wildlife, could be valuable for quantifying the transmission risk, the prevention and control, as well as the early warning, and even enhancing our ability to track wildlife-origin EIDs. Here, tissues from 32 wild gerbils of *Rhombomys opimus*, *Meriones meridianus,* and *Meiiones unguiculataus* species that were collected in the Wutonggou Desert, Fukang city, Xinjiang, were subjected to metagenomic analysis of viral nucleic acids. We selectively analyzed the relationship between certain virus families and their host animals from the perspective of viral genome evolution and explored whether there are potential viruses that may infect humans or domesticated animals.

## 2. Materials and Methods

### 2.1. Animal Sampling

After the on-the-spot investigation in the habitat (Figure 1) of Gerbillinae subfamily, 32 gerbils comprising of 3 species of *Rhombomys opimus*, *Meriones meridianus,* and *Meiiones unguiculataus* were captured by mousetraps. Due to repeated sampling in the same location, the tissues of heart (X), liver (G), spleen (P), lung (F), kidney, and bladder (SP), brain (N), intestines (duodenum, rectum, and cecum; C) of all the gerbils’ samples were combined into 7 pools according to tissue types.

### 2.2. Library Preparation and Sequencing

Given the low content of RNA virus in tissues, the samples of each pool were processed by Hipure Universal RNA Mini Kit (MAGEN, Guangzhou, China) for high purity of RNA extraction. Typically, 10~60 mg tissue was lyzed by grinding with 1 mL MagZol Reagent, followed by the addition of 200 μL of chloroform to the lysate after leaving at room temperature (RT) for 5~10 min and then shaken vigorously by hand for 15 s at RT. The samples were then centrifuged at 4 °C for 15 min at 12,000× *g* using a centrifuge (3K30, sigma, Osterode am Harz, Germany) to remove the precipitate after being left at RT for 3 min. Next, an equal volume of absolute ethanol was added to the supernatant and vortexed for 10 s. RNA was eluted by following the specific steps of the Hipure Universal RNA Mini Kit using RNAase-free water after binding the RNA to the HiPure RNA Mini Column. Double-stranded cDNA was then synthesized by reverse transcription using REPLI-g Cell WGA & WTA Kit (150,052; Qiagen, Hilden, Germany). Finally, Thermo NanoDrop One, Life Technologies Qubit 4.0, and 1.5% agarose electrophoresis were used to test the amplification products (Appendix A).

Sequencing libraries were generated using NEB Next^®^ Ultra™ DNA Library Prep Kit for Illumina^®^ (New England Biolabs, Ipswich, MA, USA) based on the manufacturer’s recommendations, and then index codes were added. Briefly, the amplified DNA was randomly sheared by ultrasound sonication (Covaris M220) to produce fragments of ≤800 bp; and sticky ends were repaired, and adapters were added using T4 polymerase (M4211, Promega, Madison, WI, USA), Klenow DNA Polymerase (KP810250, Epicentre), and T4 polynucleotide kinase (EK0031, Thermo scientific-fermentas, GlenBurnie, MD, USA). The library quality was assessed on the Qubit^®^ dsDNA HS Assay Kit (Life Technologies, Grand Island, NY, USA) and Agilent 4200 (Agilent, Santa Clara, CA, USA) system. Finally, the library was sequenced on an Illumina Novaseq 6000 (Illumina, San Diego, CA, USA) and 150 bp paired-end reads were generated. High-throughput sequencing was conducted by Magigene Company (Guangzhou, China).

### 2.3. Raw Reads Filtering and Rapid Identification of Virus Species

#### 2.3.1. Quality Control

Given that the raw data that were obtained by sequencing always include a certain proportion of low-quality reads, the Trimmomatic [45] was used here to remove the paired reads as follows; (i) with adapter, (ii) those containing low-quality base (sQ ≤ 20) over 20%, (iii) those arising from PCR duplications, as well as (iv) those with a polyX sequence to improve the accuracy of reads for follow-up analyses.

#### 2.3.2. Contamination Removal

Given the contamination of the ribosome and host sequences, all the clean reads that passed quality control were mapped to the ribosomal database (Silva.132) and host database utilizing BWA (version 0.7.17, parameter: mem –k 30) [46], respectively. Only the unmapped sequences remained for high-quality data.

#### 2.3.3. Virus Classification

The BWA [46] (v0.7.17) was used to map the high-quality reads with the GenBank non-redundant nucleotide (NT) database, and the comparison results with a length of less than 80% of the total length of the reads were filtered for high accuracy. Then, the remaining reads were classified into different virus families according to the NCBI taxonomy database annotation information.

### 2.4. Reads Assembly

The clean reads were de novo assembled by Megahit [47] (v1.1.3, parameter: --presets meta-large --min-contig-len 300), and BWA was used here to compare the clean reads with the assembly results to calculate the utilization ratio of the reads. Meanwhile, sequences that were compared to the host sequence database by BLAST (v2.9.0+) were removed. Then, Cluster Database at High Identity with Tolerance (CD-HIT) [48] (v4.7, parameter: -c 0.95 -aS 0.8) was used to cluster the assembled virus contigs of all the samples to obtain unique contigs [49].

### 2.5. Identification of Viruses

Given that viruses often carry host genes and they often have a high ribosome RNA content in the sequencing results, it is inevitable that a certain percentage of false positives will exist in the identification results. Meanwhile, a comparison based on the reference database can only identify known virus sequences. To reduce false positives and identify unknown viruses, two strategies including the identification of known viruses and the identification of de novo virus sequences, were combined in this study for more accurately identifying the virus sequences. Then, the number of phages and other viruses, as well as RNA and DNA viruses was counted based on NCBI taxonomy annotation information (2 June 2020 update).

#### 2.5.1. Identification of Known Viruses

BLAST (v2.9.0+) was used to compare the unique contigs with the virus database (Virus-NT, containing phages) that were separated from the Nucleotide Sequence Database (NT). If the alignment similarity was ≥80, the alignment length was ≥500 bp, and e ≤ 1 × 10^−5^, the contig in the comparison results would be defined as a confirmed virus sequence. At the same time, if the alignment length was ≥100 bp and e ≤ 1 × 10^−5^, the contig would be defined as a suspected virus sequence.

#### 2.5.2. Identification of De Novo Viruses

At this stage, the candidate virus sequences were searched firstly for the subsequent identification of virus sequences, using BLASTN (v2.9.0+). BLASTX (v2.9.0+) compared the unique contigs to Virus-NT and virus database (Virus-NR, containing phages) that were isolated from the Non-Redundant Protein Sequence Database (NR). The results were then filtered by satisfying e < 1 × 10^−5^ and e < 1 × 10^−3^, respectively, using MetaGeneMark (v3.38) to predict the genes and hmmsearch (v3.2.1) to compare the protein sequences with HMM (VPFs and vFam) [50,51]. The virus sequencing that met the filtering criteria (e ≤ 1 × 10^−5^) was taken as a candidate virus sequence.

The second step was to exclude the false positives. To this end, the candidate virus sequences were compared with the NT database by using BLAST (v2.9.0+) and the results were filtered to satisfy e ≤ 1 × 10^−10^. Then, a comparison was made with the previous unmatched sequences to the NR database using diamond (v0.9.10) and the results were filtered to satisfy e ≤ 1 × 10^−13^. Subsequently, the candidate virus sequences were compared with the NCBI taxonomy database; if more than 20% of the first 50 alignment results supported non-viral sequences, these alignments would be excluded, and the rest would be considered as new virus sequences [52,53].

### 2.6. Abundance Statistics of Virus

According to the comparison of the results between the virus contigs and the virus NT database by BLAST (v2.9.0+), the best hit with e < 1 × 10^−5^ was selected for species annotation, and the results without comparison were expressed by NA. Using BWA (v0.7.17, parameter: mem –k 30), the clean reads were compared to each identified viral contig and the reads with a comparison rate of less than 80% were filtered out, then the distribution of virus reads were counted according to the annotation results of virus contigs, and finally the RPKM of each viral contig was calculated.

### 2.7. Prediction of Gene

Prokka [54] (v1.13) was used to predict the gene sequences of the virus contigs followed by filtering the contigs with a length of less than 200 bp. The predicted gene number, length, etc. were evaluated for metagenomic analysis.

### 2.8. Phylogenetic Analysis

The whole-genome sequences of virus strains, the same species as the dominant viruses in gerbils from different hosts were downloaded from ViPR database and NCBI database. Sequence alignment was performed using MAFFT [55] (v7.487) with the auto-alignment strategy. The best substitution models, as well as the maximum likelihood (ML) trees, were then performed by MEGA X [56] (version: 10.2.6) with 1000 bootstrap replicates.

## 3. Results

### 3.1. Metagenomic Analysis and Virome Overview

A total of seven libraries were constructed according to tissue types and deep sequenced by high-throughput sequencing. We obtained 147.6 GB of data (492,325,915 valid reads with an average length of 150-bp were generated) and a total of 365,838,486 effective reads were employed for further analysis after the removal of low-quality sequences in the raw reads. Of these, 136,519 reads were best matched with viral proteins that were available in the GenBank non-redundant database (0.04% of the total effective reads), with the number of virus-related reads in each library varying from 6506 to 36,444 (Table 1) and each library having a sequence match to the virus.

### 3.2. Analysis of the Results of Species Taxonomic Annotations

High-throughput sequencing revealed the genes that were involved in 38 virus families and some unclassified families (Appendix A), where the most widely distributed virus families were *Flaviviridae*, *Retroviridae*, *Nairoviridae*, *Myoviridae*, and *Herpesviridae* (Figure 2), and the virus reads that were associated with these families accounted for 88.86% of the total virus sequences by viral metagenomic profiling. These viral reads, derived from animals, insects, plants, and bacteria, revealed a large degree of viral diversity. The virus families that can infect vertebrate and invertebrate hosts simultaneously, such as *Reoviridae*, *Togaviridae*, *Iridoviridae*, etc., are potential vector-borne virus families and were detected more commonly in the heart, brain, kidney, and bladder arrive viscera pools.

### 3.3. Analysis of Sequence Assembly and Gene Prediction

A total of 3,649,734 contigs of variable length were then yielded for further viral metagenomic composition analysis by de novo assembly using Megahit. All the contigs of each sample were clustered using CD-HIT, resulting in 3,428,363 unique contigs (Appendix A) with a maximum length of 20,718 bp (Table 1). At this stage, the unique contigs that were longer than 100 bp were then classified using BLAST as seven confirmed viral contigs (Table 2) and 90,246 suspected viral contigs (Appendix A) based on the taxonomic origin in the annotation of the best-hit sequence (e-value < 10^−5^) with the GenBank non-redundant database. Then, an assignment of these contigs to a combination of multiple databases identified a total of 5069 virus contigs with 51.39% DNA viruses and 48.61% RNA viruses, among which 7.05% were assigned to phages (virus-NT virus contigs without matching results that are temporarily unclassified, and, therefore, are not within the statistics).

In subsequent analysis, the contigs that were identified as DNA viruses in the above two strategies were removed, while the contigs that were identified as RNA viruses and contigs that were not known to be RNA or DNA viruses were retained, respectively. We then compared the difference between RNA viruses and the unknown virus contigs that were obtained by the two strategies and retained the viral contigs in the novel and confirmed final viruses (Appendix A). In the aggregate, a total of 1848 contigs (Appendix A) in the novel and confirmed were retained as the result of the identification of the final virus sequences. The analysis of all the viral sequences revealed that these sequences could be assigned to 19 different viral families. The most common viral sequences were assigned to the viruses infecting mammals, followed by the viruses that can infect various insects or plants and bacteria or fungi reflecting the large insect and plant rodent diet, as well as those whose natural host range remains undefined (Appendix A and Figure 3).

About 98.92% (*n* = 1828) of sequences were annotated for the vertebrate virus families, including a total of 15 virus families. The most common taxonomical family was *Retroviridae*, which accounted for a maximum of 78.79% in vertebrate viruses. This was followed by the *Flaviviridae* of 7.52%. Of note, *Pneumo-*, *Picobirna-,* and *Nairoviridae* each contributed between 1 and 5% of the sequences that were attributed to vertebrate-infecting viruses, whereas the other important vertebrate pathogen viral families such as *Arena*-, *Hepe*-, *Phenui*-, *Rhabdo*-, *Calici*-, *Reo*-, *Corona*-, *Orthomyxo*-, *Peribunya*-, and *Picornaviridae* each contributed less than 1% of the sequences that were attributed to vertebrate-infecting viruses. At the same time, *Alphaflexi-*, *Chryso-*, *Partiti-,* and *Tospoviridae* represented the majority of plants and algae virus sequences. A total of 0.11% of viral sequences were without any assignment to a known viral taxon after BLAST analysis (Appendix A).

Based on the comparison results between the virus contigs and the virus-NT database, the results were annotated by selecting the best hit (e < 1 × 10^−5^) for all the remaining unique contigs. After remapping the clean reads and calculating the RPKM of each virus contig, members of the families *Flaviviridae*, *Arenaviridae,* and *Retroviridae* represented the most abundant species, and only the top 30 were shown here (Figure 4). A total of 772 contigs were then predicted as ORF, where the genes were aligned through the BLAST software and NCBI virus database by Prokka, according to the number of predicted genes in each contig, the number of contigs with only one gene (649), two genes (38), and more than three genes (12) were counted, respectively.

### 3.4. Interesting Taxonomical Families of Vertebrate Viruses due to Zoonotic Potential

#### 3.4.1. Retroviridae

Endogenous retrovirus sequences are a common presence in the genomes of almost all animals, including fish, amphibians, reptiles, birds, and mammals. Here, the 1456 contigs identified as members of the Retroviridae family from seven libraries were assigned to three major genera: *Betaretrovirus*, *Grammaretrovirus*, and *Lentivirus*. The species-level taxonomical annotation indicating the presence of human retroviruses, such as human mammary tumor retrovirus, human endogenous retroviruses H, K, and W, etc., the matching sequences exhibited 63–100% identities at alignment lengths of 100–1381 bp. The most interesting were the sequences with a 5367-nucleotide-acid and a 4025-nucleotide-acid length that indicated the presence of Moloney murine sarcoma virus (M-MSV) and Murine leukemia virus (MuLV), at similarity values of 76.19% and 66.59%, respectively.

#### 3.4.2. Flaviviridae

The *Flaviviridae*, a family of enveloped positive-strand RNA viruses which mainly infects mammals and is primarily spread through arthropod vectors (mainly ticks and mosquitoes), is the next most abundantly identified group of vertebrate viruses with zoonotic potential. There were 139 contigs that were identified in *Flaviviridae* distribution among four genera: *Flavivirus*, *Pestivirus*, *Pegivirus*, and *Hepacivirus*, and the sequences of Heciviruses (HVs) made up nearly half of the contigs. All the alignments indicated between 66.37% and 93.44% identities to the database reference viruses and at alignment lengths between 100–6598 bp. The relatively long rodent *Hepacivirus*-like contigs with the alignment length of 6598 bp, 3277 bp, and 2840 bp were isolated from the heart, liver, and lung viscera pools, and these contigs shared 71.85%, 72.90%, and 69.85% homology with rodent Hepaciviruses that was isolated from rodents (GenBank accession: KY370092.1, 8989 bp), respectively. The longest contig that was isolated from the heart accounted for 73.40% of the total reference genome. This contig has high homology with the RdRp sequence of the reference sequence. It could have originated from the same virus strain as the one that was isolated from rodents in Inner Mongolia province, China in 2018. Other contigs with a high consistency with rodent Hepacivirus ranged in length from 105 bp to 1878 bp (Appendix A).

#### 3.4.3. Nairoviridae

Among the *Nairoviridae*-like sequences, we found indications of the presence of *Thiafora-*, *Chim-*, *Dugbe-*, *Hughes-*, and Nairobi sheep disease orthonairovirus, except for two sequences resembling *Thiafora orthonairovirus* that aligned at relatively high similarities of 64.36% and 68.42% at alignment lengths of 1341 and 1067 bp, respectively. Others with short alignment lengths showed various levels of similarity.

#### 3.4.4. Picobirnaviridae

Picobirnaviruses (PBVs) are small, non-enveloped, bisegmented, double-stranded RNA viruses that have been frequently detected in mammals, birds, invertebrates, and environmental samples which could be implicated in gastroenteritis in animals and humans, but the disease association is unclear. Interestingly, in pool C, we found indices of various species-level taxa including porcine-, feline-, rat-, marmot-, murine-, human-, goat-, rabbit-, mongoose-, and chicken *picobirnavirus*, as well as *Picobirnavirus* sp. among contigs of *Picobirnaviridae* (Appendix A). The dataset suggested that there were two viral sequences of marmot picobirnavirus and one of goat picobirnavirus, highlighted by the presence of putative protein and the capsid protein, at similarity values of 66.22%, 70.19%, and 69.71%. While the first of three sequences aligned to segment I of PBVs, the latter two viral sequences related to porcine-, feline-, as well as two that related to marmot picobirnavirus at similarity values of 82.67%, 68.43%, 69.89%, and 70.98% to the viral RdRp sequence at alignment lengths of 1639 bp, 1080 bp, 1023 bp, and 1137 bp, respectively. Additionally, two viral sequences resembling *Picobirnavirus* sp. aligned at similarities of 73.29% and 77.02% to the viral RdRp at alignment lengths of 1591 and 1079 bp, respectively, hinting relatedness to segment II of PBVs.

#### 3.4.5. Low-abundance viruses

Viral sequences that were related to *Pneumo-*, *Arena-*, *Phenui-*, *Calici-*, *Hepe-*, *Orthomyxo-*, *Peribunya-*, *Rhabdo-*, *Corona-*, *Picorna-*, and *Reoviridae*-like were at short alignment lengths at variable alignments similarities. Moreover, all *Pneumoviridae*-like sequences resembled Human orthopneumovirus at various similarity rates, ranging from 65.78 to 76.85%. We found an indication of only *Alphacoronavirus* sp.-like sequence, although at a similarity level of 78.22% and the alignment length of 34 amino acids. All five *Rhabdoviridae*-like sequences that resembled *Xingshan alphanemrhavirus* [52] indicated the presence of mammalian rhabdoviruses, and one of them contained a 132-amino-acid sequences coding for amino acids in protein (CDS) at the similarity of 71.88% to the hypothetical protein. The presence of *Manzanilla orthobunyavirus* might have been suggested by the detection of the single *Peribunyaviridae* sequence and the *Reoviridae* sequences indicated the presence of *Rotavirus A* virus.

### 3.5. Ten Genome Sequences of Novel Viruses Were Identified

After de novo assembly of the read pairs, we identified and characterized 10 nearly complete novel genome sequences, including a rodent *Hepacivirus*-associated virus (GenBank accession No.: OM179960), and nine PBVs, comprising three segment I and six segment II sequences (GenBank accession Nos.: OM142648–OM142656).

#### 3.5.1. Flaviviridae

The Hepacivirus genome consisted of non-segmented, single-stranded, (+) sense RNA, approximately 10 kb long, which encodes a single ORF that is translated into a polyprotein. Whole or partial genome genes of four genera in *Flaviviridae* from different hosts including humans, bats, rodents, and other mammals were obtained from the NCBI database (National Center for Biotechnology Information (nih.gov)). After sequence alignment was conducted by ClustalW of MEGA X, the best substitution model analyzed by MEGA MODELS was GTR+G+I, then the tree was constructed using a maximum likelihood method and a bootstrap analysis with 1000 trials was performed. The phylogenetic analysis revealed the closest relationship between the 8508 bp length sequence and rodent Hepacivirus that was isolated from rodents (KY370092.1) [17]. They all cluster in the Hepacivirus of the *Flaviviridae* family (Figure 5).

#### 3.5.2. Picobirnaviridae

The genome of PBVs consisted of two segments. The large segment contained two open reading frames encoding a polyprotein (ORF1) that self-cleaves to yield the mature coat protein and a large peptide and the capsid protein (ORF2), respectively; whereas the small genomic segment contained a single ORF which encodes for the viral RNA-dependent RNA polymerase (RdRp) [57]. Based on the RdRp gene, PBVs are classified into different genogroups. A total of nine new genotype viral sequences, comprising three segment I and six segment II sequences, were identified in the intestines and liver of seven viscera pools. In the two phylogenetic trees that were constructed using ORF1 and ORF2 of segment I, the three segment I sequences were clustered into distinct clades. Phylogenetic analysis showed that the PBV sequences were distributed in the whole tree that was constructed using the RdRp gene of segment II with high diversity, sharing 68.42–82.67% nucleotide identities with other genogroup I and genogroup II PBV strains based on the partial RdRp gene (Figure 6).

## 4. Discussion

Emerging zoonotic diseases have received tremendous interest in recent years. With the advent of advanced molecular technologies such as high-throughput sequencing, meta-genomics, meta-transcriptomics, etc., the virome studies under viral metagenomics has formed a relatively new branch of virus genetic research. This new research area enables a systematic analysis of the existence of the vast majority pathogenic or non-pathogenic viruses, known or unknown viruses, and other endogenous or exogenous viruses that are carried by the wildlife, and enables the exploration of the roles of wild species as reservoirs of infectious diseases based on the results of Next Generation Sequencing (NGS) [58,59,60]. As rodents are the largest mammalian population with diverse and widespread groups that are important reservoirs of many zoonotic viruses that have large impacts on humans and other animals’ health, interest in viral diversity in rodents has recently increased [13,17,37,61,62,63]. Indeed, approximately 173 viral species belonging to more than 65 genera have been described in rodents to date, among which about 60 are zoonotic, such as *hantaviruses* (*Bunyaviridae*), *mammarenaviruses* (*Arenaviridae*), and *picornaviruses* (*Picornaviridae*) [13,64,65,66]. Therefore, investigating the potential rodent-borne viruses should be an important part of the early warning and traceability of infectious diseases and emerging infectious. In addition, further analysis of rodent-borne viruses by viral metagenomics not only will increase our understanding of the diversity of viruses that are carried by wild rodents, but also will reveal that a significant number of rodent viruses are still unknown.

Disturbance levels and habitat types also affect the viral diversity index and the level of comparative advantage of viral species. To analyze the viral infection status in wild gerbils and to identify viral genomes at the core RNA virome in this study, the heart, liver, spleen, lung, kidney, bladder, brain, and intestines representing different tropisms of viruses of all gerbils’ samples were combined into seven viscera pools according to tissue types for analysis. To reduce false positives and identify unknown viruses, two strategies of homologous sequences and de novo virus sequences were combined to identify the viral sequences. Overall, a total of 22 viral families that were closely related or distantly related to known viral sequences and the sequences of many unknown taxonomic origins were identified. Viruses vary in families from those of *Arenaviridae*, *Retroviridae*, *Flaviviridae*, *Nairoviridae*, and *Picobirnaviridae*, etc., that are associated with vertebrates, to those of potential vector-borne viruses such as *Togaviridae*, *Reoviridae*, and *Rhabdoviridae*, which all have been detected in the seven viscera pools. On the other hand, different viral families, whether they are originating from animals, plants, or bacteria, were not evenly distributed within the different tissues, such as viruses of *Picobirnaviridae*, *Arenaviridae*, and *Flaviviridae,* except for *Retroviridae* which was mostly detected in the different tissues of gerbils. Although members of distinct viral families, most RNA viruses give a good representation of vertebrates and invertebrates; even ameba and plants viruses reflect the mixed diets of rodents.

Many members of *Flaviviridae* can infect humans and mammals. Diseases that are associated with *Flaviviridae* include *hepatitis* (*hepaciviruses*), hemorrhagic syndromes, fatal mucosal disease (*pestiviruses*), hemorrhagic fever, encephalitis, and the birth defect microcephaly (*flaviviruses*). Although there is a distinct evolutionary difference among human Hepatitis C virus (HCV) and other Hepacivirus (Figure 5), the Hepacivirus genera, once considered an isolated infection in humans and other primates, has now expanded to include horses [26], rodents [17,27,28], bats [29], cows [30], and other wildlife. All these further reveal the wide host adaptation and may reflect the frequent cross-species transmission of Hepacivirus rather than diversification from a common source. Cross-species transmission of viruses often leads to accelerated viral evolution and altered virulence patterns, and the mutated strains of COVID-19 have illustrated this point. Meanwhile, as more and more hosts of HCV are discovered, new clues to the origin of HCV may be provided. The Hepacivirus genus consists of two species: HCV and GB virus (GBV) [67]. The GBV is much more evolutionarily similar to rodent HCV than human HCV (Figure 5) [28]. The HCV that are found in rodents, bats, dogs, horses, and other wildlife are tentatively known as non-primate HCV viruses (NPHVs) [29]. Rodent *Hepacivirus*, which we identified here, shows the highest amino acid or nucleotide homology to Rodent Hepacivirus isolate RtMm-HCV/IM2014 that was previously reported to infect rodents from Inner Mongolia, China [17]. This virus has also been detected in wild gerbils from Wutonggou Desert, and this is also the first time that HCV has been detected in rodents in Xinjiang, China, confirming the presence of HCV in rodents in China. These two strains of hepatitis C virus were isolated from rodents in Inner Mongolia and Xinjiang, respectively. While Xinjiang and Inner Mongolia, the latter lying in northeast China, has a similar ecological environment with an abundance of rodent species, our study shows that rodents under this ecology could be the potential carriers of these genus viruses.

PBVs have been detected in the feces of humans and a wide range of animal species with or without diarrhea [68,69,70,71,72], and about 20% of human fecal diarrhea samples of unknown etiology were positive for PBV in the Netherlands [40]. The viral sequences of PBV in this study matched two different virus segments that include segment I and segment II. These sequences were highly divergent from other *Picobi**rnaviruses* for which full-length or segment-length sequences were available, especially in the capsid-coding segment I. Segment II, which encodes the RdRp, was more conserved but still displayed a high degree of diversity with PBV sequences from many other host species. Phylogenetic analysis of the nearly complete sequence of segment II, for which 6 *Picobirnavirus* sequences were available, indicated that rodent *Picobirnaviruses* were highly diverse, belonged to two different phylogenetic clades (Figure 6), and were grouped with porcine, marmot, murine, and gorilla *Picobirnaviruses*. Our data further corroborate previous reports on the high genetic diversity of *Picobirnaviruses* [69,73,74]. Unfortunately, the pathogenicity of *Picobirnaviruses* has not been established.

This study detailed the core RNA viromes that reside in different tissues of wild rodents. Among the different viral families that were detected here, *Retroviridae*, *Nairoviridae*, *Flaviviridae*, and *Picobirnaviridae* have been described as major zoonotic virus families. There have also been findings of low-abundance but highly contagious zoonotic viruses, such as viral sequences related to *Pneumo-*, *Arena-*, *Phenui-*, *Calici-*, *Hepe-*, *Orthomyxo-*, *Peribunya-*, *Rhabdo-*, *Corona-*, and *Picornaviridae* that are harbored by rodents, which indicated the diversity of viruses that are carried by rodents as the natural host of zoonotic viruses. Although the data showed that only a single contig was annotated as coronavirus, this could not confirm the existence of coronavirus in rodents. It may be due to the limitations of virus metagenomics technology that we cannot thoroughly know the diversity and richness of viruses that are carried by host animals. However, some studies have shown that rodents may serve as intermediate hosts of the coronavirus [75], which suggests that we need to further explore the complex viral ecosphere among virus hosts and discover the source and mode of transmission of zoonotic viruses.

Rodents can serve as a potential reservoir for many animal-derived viruses which can exist stably in the natural hosts and do not cause clinical symptoms. However, when these viruses break the species barrier to infect humans, they often pose a great threat to human health. Nearly 70% or more of the viruses that cause new infectious diseases are of animal origin [2] and their genome sequences differ greatly from those of known pathogens. As shown, there are many rodent virus species and genera that were identified in this study, most of them were related to viral sequences that were previously detected in rodents worldwide. Hence, the identification of animal viruses is important for understanding the animal disease, the origin, and evolution of human viruses, as well as zoonotic reservoirs for emerging infections. The epidemiological characteristics of viruses, molecular epidemiological trends, distribution, and diversity of viruses and hosts still need to be studied, which will certainly have a positive effect on human response to emerging outbreaks of infectious diseases of animal origin in the future. Further studies are needed to investigate the ability of rodents as natural hosts to transmit the viruses and to elucidate the true impact of zoonotic viral infections on human welfare.

## Figures and Tables

**Figure 1 viruses-14-00356-f001:**
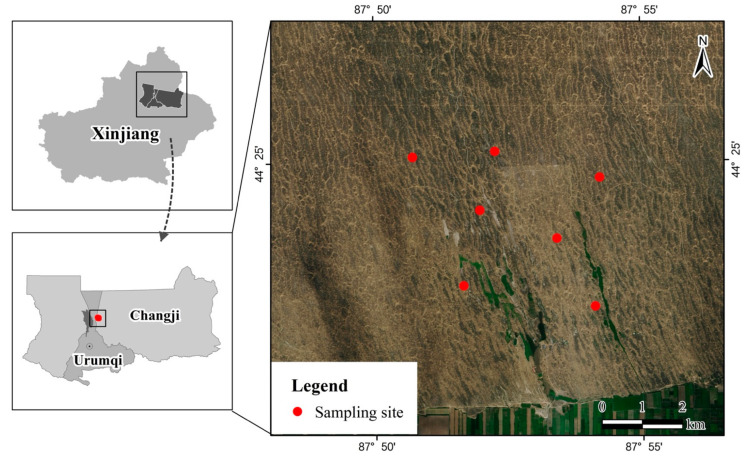
Locations of rodent sampling in the Wutonggou Desert, Fukang city, Xinjiang China.

**Figure 2 viruses-14-00356-f002:**
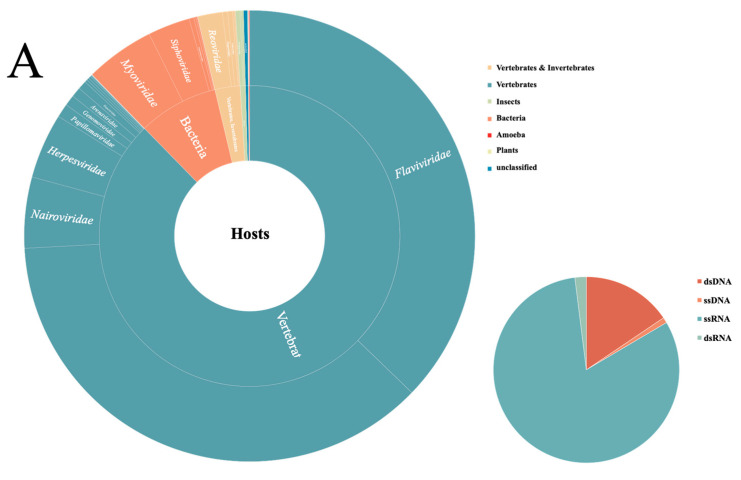
(**A**) Overview of the viral reads. The numbers of viral reads from each sample are described in Additional file Appendix A. (**B**) The percentage of reads that are related to the most abundant viral families among all the virus reads, indicated in the same colors for each main viral category. virus families are indicated by the color code on the right.

**Figure 3 viruses-14-00356-f003:**
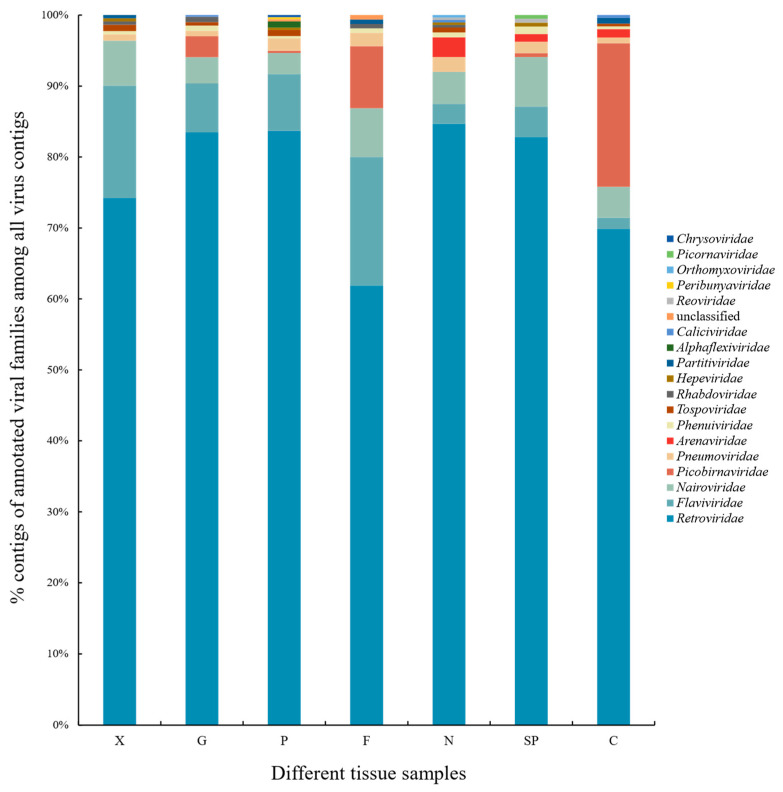
Histogram of virus contigs distribution in different tissue samples of gerbils.

**Figure 4 viruses-14-00356-f004:**
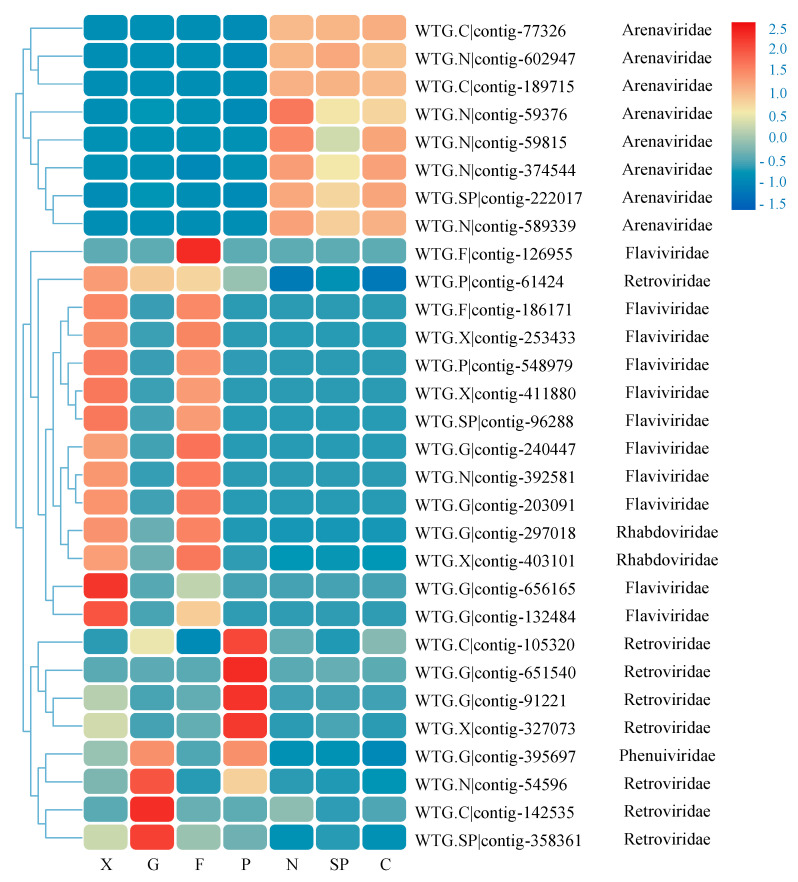
Heatmap of contigs with the top 30 abundance of sequence reads in each sample. The tissue types are listed below the heatmap. Information of contigs and the virus families they belong to is provided in the right text column. The boxes that are colored from blue to red represent the abundance of virus reads aligned to each contig.

**Figure 5 viruses-14-00356-f005:**
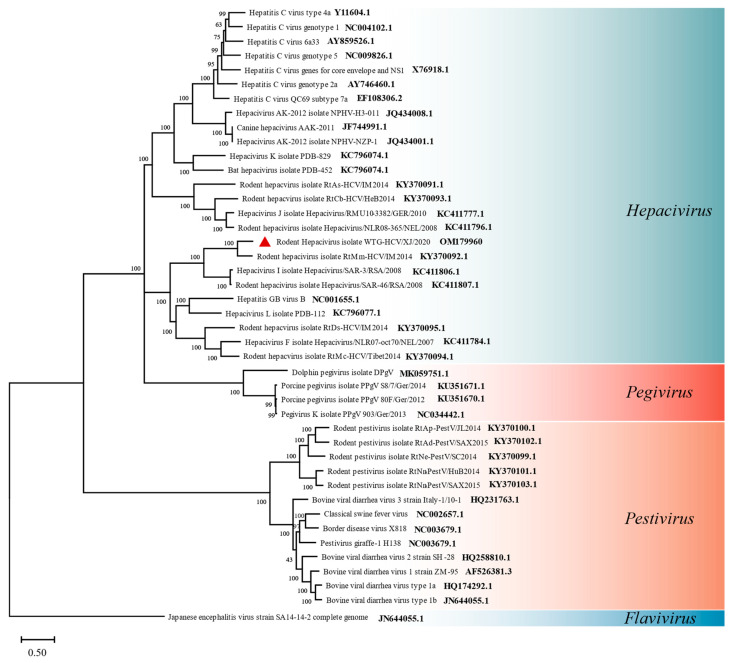
Phylogenetic relationship based on 8772 bp long partial polyprotein region (217–8988 bp on polyprotein of rodent Hepacivirus isolate RtMm-HCV/IM2014 polyprotein gene, acc.no.KY370092) of Hepacivirus and other members of *Flaviviridae* from rodent and other hosts. The analysis was inferred using the maximum likelihood method based on MEGA X. GenBank accession numbers are indicated at the branches of the tree, if available. Branch bootstrap values are shown and were based on 1000 bootstrap simulations. The contig of rodent Hepacivirus from gerbils is marked with a red triangle. The scale bar depicts an evolutionary distance of 0.50.

**Figure 6 viruses-14-00356-f006:**
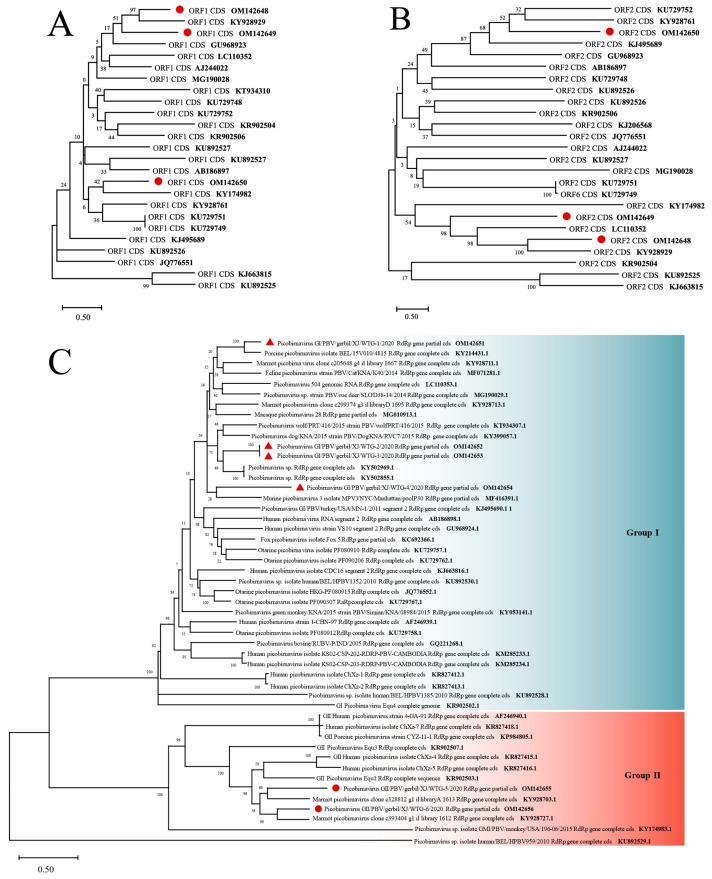
Maximum likelihood phylogenetic tree of Picobirnaviruses based on (**A**) ORF1; (**B**) ORF2; and (**C**) viral RNA-dependent RNA polymerase sequences of segment 2. GenBank accession numbers are followed by virus names. There were three closely related variants of PBVs (82.67%, 73.29%, 77.02% nucleotide identity in RdRp gene respectively) that were identified in the present study and are marked with a red triangle. The numbers beside the branches represent statistical confidence in clades based on 1000 bootstrap replicates, only bootstrap values ≥ 50% are shown. Scale bar (0.50) = nucleotide substitutions per site.

**Table 1 viruses-14-00356-t001:** Overview of reads and contigs sequences of various tissues from gerbils.

Samples	Raw Reads	Number of Reads Remaining after Filtering (%)						Assembly Data on Rm. rRNA Clean Reads				
		Clean Reads (PE)		Rm. rRNA (PE)		Virus Reads (PE)		Total No.	Max Len.	Min Len.	N50	GC (%)
C	68,016,775	53,477,225	78.62	27,236,851	50.93	6506	0.01	411,219	12,701	300	610	45.46
F	71,127,692	55,001,720	77.33	52,231,458	94.96	36,444	0.06	261,682	4443	300	512	45.61
G	71,699,768	54,411,245	75.89	53,262,395	97.89	25,760	0.05	656,296	9489	300	543	46.26
N	71,058,840	51,163,121	72	50,711,357	99.12	8278	0.02	708,127	18,386	300	866	43.9
P	70,803,346	52,148,081	73.65	52,071,866	99.85	27,241	0.05	680,681	20,718	300	741	44.7
SP	70,587,365	47,041,404	66.64	46,790,577	99.47	11,248	0.02	504,349	8466	300	584	43.92
X	69,032,129	52,595,690	76.19	51,802,597	98.49	21,042	0.04	427,380	6831	300	527	45.61
Mean	70,332,274	52,262,641	74.33	47,729,586	91.53	19,503	0.04	521,391	11,576	300	626	45.07
SD	1,314,630	2,651,684	0.04	9,274,055	0.18	11,191	0.02	166,912	6037	0	131	0.91

Rm. rRNA clean: number and percentage of reads after removing ribosome sequence; Virus reads: number of reads mapped to the virus database; SD: standard deviation; PE: paired-end.

**Table 2 viruses-14-00356-t002:** Seven contigs that were confirmed for virus species.

Query Id	Subject Id	Identity (%)	Alignment Length	e-Value	Bit Score
WTG.F|contig_243821	LR597639.1	100	585	0	1056
WTG.X|contig_22842	HQ540595.1	82.765	586	2.98 × 10^−169^	602
WTG.X|contig_63540	M87581.1	99.592	1226	0	2189
WTG.C|contig_132444	MK032740.1	83.309	2037	0	2118
WTG.C|contig_85266	KY214431.1	82.672	1639	0	1629
WTG.G|contig_109493	KY214431.1	85.562	658	0	751
WTG.C|contig_323805	MH412924.1	84.261	521	1.33 × 10^−155^	556

## Data Availability

The data presented in this study are available in this article and on request from the corresponding author.

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
