# Peer review of "Metagenome-Assembled Viral Genomes Analysis Reveals Diversity and Infectivity of the RNA Virome of Gerbillinae Species"

_viruses, 2022, doi:10.3390/v14020356_

Round 1

Reviewer 1 Report

page 11 figure 5 : confusion between hepacvirus and hepacivirus in the figure (5 hepacvirus) and in the text (ex: rodent hepacvirus with red triangle in the text (ligne 324) and  rodent hepacivirus in the figure for KY370092.1, also page 13, ligne 361

ligne 444 page 15: picobirnaviruses and not picobinaviruses

Reviewer 2 Report

 Analysis of metagenome-assembled viral genomes from the rodents reveals diversity and infectivity of the RNA virome, by Han Du et al.

In this study, the authors searched for the presence of viral sequences in different organs of 32 gerbils (3 different species) from Xinjiang. It is of interest to understand more upon rodent virome to determine the zoonose potential of those viruses carried out by rodent or other wildlife. Many works deal with this subject, both in rodent and in bats.

In the present study, the authors have chosen to pool their samples according to the organs tested. Thus, they detected 1848 viral genomes representatives of different viral families or genus.

The study is conducted seriously but I have a few comments. Indeed, many metagenomic works on rodents are currently published, and sometimes the number of samples processed is much larger. Here only 32 animals were tested and in a small geographical area (a few km2).

The authors remained very descriptive in this work and have not carried out any additional validation of their results. We could hope to have at least one validation data on the long Rodent hepacivirus-like contig, using PCR amplification on individual samples to have an idea on the prevalence of those sequence in each individual and/or organ. This type of results could give greater depth to this work.

The number of individuals tested (32 animals) remains limited to a small geographical area and this validation could be considered for a wider area in order to verify whether these data are really specific to this region or, on the contrary, representative of a wider epidemiology. Other species of rodent could be tested.

Minor comments:

  • de novo to be homogenized in the text, sometimes denovo and other de novo
  • line 138: often have
  • line 147-150 : not clear
  • line 152: At this stage
  • line 152-158 : clarifiy virus-NT and virus-NR ; perhaps you need to add abbreviation table.
  • Line 210 : CDHIT add the total name
  • Table 1: reverse number and percentage in the legend. Define PE.

Reviewer 3 Report

Dear authors,

in the file attached you find the needed revisions.

Best,

the reviewer
